# Do the Culturable Microbial Groups Present in Cutaway Bogs Change According to Temporal Variation? Pilot Study Based on the Midlands in the Republic of Ireland

**Gouri Nilakshika Atapattu** [1,2][ID]**, Tara Battersby** [2]**, Michelle Giltrap** [1,3] **and Furong Tian** [1,3,*][ID]

1   School of Food Science Environmental Health, Technological University Dublin, City Campus, Grangegorman, D07ADY7 Dublin, Ireland
2   Environmental Sustainability and Health Institute, Technological University Dublin, City Campus, Grangegorman, D07ADY7 Dublin, Ireland
3   FOCAS Research Institute, Technological University Dublin, City Campus, Camden Row, D08CKP1 Dublin, Ireland
*   Correspondence: furong.tian@tudublin.ie

**Abstract:** Cutaway peatlands in the midlands of the Republic of Ireland are rarely the focus of scientific studies. The soil quality and related microenvironment have been severely impacted by peat extraction. Returning them to a 'near-natural state' would require greater insights into this ecological niche. The current research took the initiative to study the microbiology of vast cutaway sites in the midlands of Ireland. Peat was collected over two seasons in January, February and April. Homogenised peat was aseptically cultured on a range of specific and non-specific culture media. Microbial enumeration, Gram staining and other microscopic observations of morphologically distinct microorganisms were performed. The total viable bacterial and fungal numbers were highest in February ($1.33 \times 10^5$ CFU ml$^{-1}$ and $5.93 \times 10^6$ CFU ml$^{-1}$, respectively) and were lowest in April ($1.14 \times 10^3$ CFU ml$^{-1}$ and $5.57 \times 10^6$ CFU ml$^{-1}$). *Penicillium* spp. and *Trichoderma* spp. were common in all the sites. The highest values of phosphate solubilisation index were recorded in peat collected in April (SI = 3.167 & 3.000). Overall, there is a statistically significant difference ($p \leq 0.0001$) among the microbial numbers across the three months. This variation could be due to the temperature and pH differences across peat soil.

**Keywords:** cutaway peatlands; aerobes; fungi; anaerobes; actinomycetes; restoration

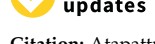



## 1. Introduction

Peatlands are water-logged ecosystems that can store thousands of years' worth of carbon in the depths of their soil [1]. In order to gain land, modern methods strip surface vegetation away and dry out the surface layers using large machines [1]. During the peat extraction process, the original surface peat growing vegetation is disrupted. It eventually leads to peat drainage [2]. Drainage-based practices in agriculture and forestry have caused approximately 15% of peatlands to become degraded worldwide [3]. The long-term effects of peat harvesting include peat compaction, formation of different vegetation types, manipulation of soil chemistry and erosion [1,2]. Furthermore, hydrological functions are impaired by this process, which drops the water table level [4]. The combination of low water table level and high-oxygen content can facilitate accelerated rates of microbial decomposition and subsequently cause the $CO_2$ gas emissions to increase [4,5]. When there is no longer an economic supply of peat at a particular site, it becomes rich in atmospheric carbon [2,6]. This leads to continuous greenhouse gas emissions [6].

Considering the major impacts of peatland drainage, the Republic of Ireland has launched several restoration projects across the country. One primary example is the 'Rehabilitation of cutaway peatlands in the midlands of Ireland'. Its objective was to recreate

wetland nature in previously used industrial cutaway bogs [2]. The BOGFOR research program (1998–2005) was another Irish project. BOGFOR addressed the key components in the afforestation of industrial cutaway lands. The main challenges were soil heterogeneity and avoiding the dominance of competitor plant species [7]. Introducing peat-forming plant species into the land has received the most attention in peatland restoration practices [8]. Plant community studies have been used widely for restoration, while there is a paucity of studies analysing the soil microbiology of cutaway peatland ecosystems. Microorganisms can regulate many interactions between plants and soil [9]. A proper understanding of the 'terrestrial carbon cycle' is required to implement novel restoration practices because the soil microbial community greatly impacts the carbon cycle. Microorganisms mediate the major steps of the terrestrial carbon cycle [10]. However, no studies exist on different types of microbial groups in relation to numbers of and temporal variation in cutaway bogs. Information about peat soil in the scope of microbiology has the potential to unravel unseen dynamics of the peat environment. Novel findings can introduce effective re-wetting schemes. Urbanová and Bárta assumed that methanogenic microorganisms in peat can act as indicators in the observation of environmental conditions [11]. The effect of re-wetting schemes can be monitored by the numerous yet reliable pieces of information obtained from assessing the methanogenic Archaea present in peat. In conclusion, the study emphasised the importance of investigating the anaerobic microbial populations [11]. One recent study reported the use of bioindicator values of 'mites and vegetation' to assess the quality of peatlands. They observed significant changes in the number of species of vegetation present in recovering peatlands [12]. Likewise, the microbial numbers and communities in a drained peatland could be different than its pristine conditions (the pristine condition of a peatland is its original state and often refers to a 'near-natural' condition as reference). However, not only methanogens but other groups of microorganisms are also pivotal in understanding the necessary steps of reclamation since they mediate certain reactions in the terrestrial carbon cycle. These data can be employed as bio-indicators to monitor the restoration success of drained cutaway sites. It is necessary to monitor the activities of a peatland ecosystem against baseline or reference data to assess the restoration progress. The availability of these data is inadequate in Western Europe [13]. Throughout restoration, the drained lands reach 'near-pristine conditions'. Culture isolation is one way to find indicator organisms. While this is a long-term goal, our pilot study took the initiative to collect the necessary baseline data about culturable populations to reach that goal in the future. If studies are focused on microbial groups in the sense of modifying them as microbial/environmental indicators, the complexity of the restoration process can be gradually minimised. Considering this 'potential' of microbes, our study presents the following aims and objectives for this pilot research, i.e., (i) to collect baseline data of culturable microbial groups present in each cutaway site, (ii) to compare the microbial numbers according to different groups in each site and (iii) to demonstrate the possibility that microbial communities can act as an environmental measure for peatland restoration. The soil–microbial ecosystem must be best understood in the restoration process as it cannot not be achieved in a single day.

This is the first time in Ireland a pilot microbiological study of this type has been conducted based on cutaway peatlands in the midlands (midland accounts for the main production in cutaway peatlands) [6]. The current study collected preliminary data of six different microbial groups according to temporal variation. The actual total microbial numbers might be higher than the ones illustrated (through culture isolation) in this paper.

## 2. Materials and Methods

### 2.1. Experimental Set Up and Study Sites

Samples were collected from study sites in County Offaly (53°9′46″ N 7°39′10″ W), County Laois (53°9′49″ N 7°37′18″ W) and County Tipperary, located in the midland of the Republic of Ireland (Figure 1). Peat across three cut- away-rough-grazing sites was collected over winter and spring seasons (January, February and April) in the year 2022. The purpose

of this experiment setting was to compare peat microbiota across the cutaway peatland sites and determine if there were significant differences in the tested microbial parameters.

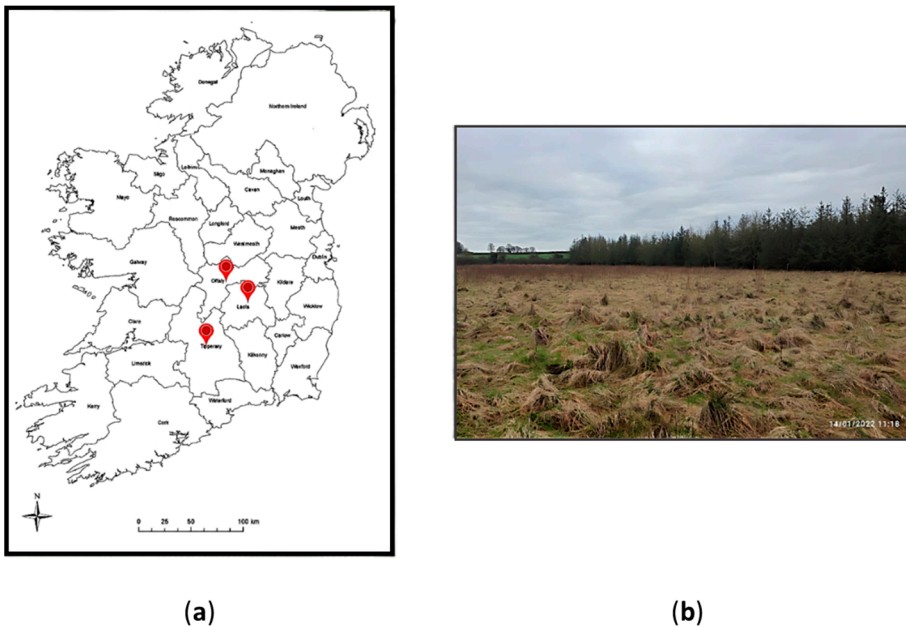

(**a**)                                                                                           (**b**)

**Figure 1.** (**a**) A map of Republic of Ireland denoting the sampling locations in the midlands; (**b**) The geographical view of the cutaway peatland located in County Offaly on January 2022. (The average temperature was 6 °C).

### 2.2. Sampling

The vegetation type of the peat was carefully observed before the sampling process was undertaken. Some of the observed vegetation types were *Sphagnum* mosses and sedges, which the lands were dominated with. Each site was divided into 10 strips with the form of zigzags. Ten composite soil samples were collected from each site in order to represent the whole site. Replicating the sample size would result in minimum errors. The depth of the soil was in the range from 0 to 15 cm. A standard soil knife was used to dig the soil into relevant depths. Peat soil 250–300 g was collected in previously sterilised airtight containers. The samples were kept in around 4–6 °C.

### 2.3. Isolation of Different Types of Culturable Microorganisms

#### 2.3.1. Isolation and Enumeration of the Total Viable Bacteria and Aerobic Bacteria

The soil samples were sieved through a 2 mm sieve to remove small stones, fauna and plant debris. Fresh peat (10.0 g) was suspended in 90.0 mL of sterile maximum recovery diluent. It was dispersed in a homogeniser at 140 rpm for 1 h. The resulting suspension was diluted by serial, 10-fold stages. The pour plate technique (1ml aliquots) was performed for the dilutions $10^{-1}$–$10^{-7}$ on a nutrient agar for the detection of the total viable bacteria. The pH of the medium was $7.0 \pm 0.2$. Likewise, the serial dilutions were spread-plated (0.1 mL aliquots) onto nutrient agar and tryptic soy agar (TSA) plates for the detection of aerobic bacteria. The plates were incubated in inverted positions at conditions of $37 \pm 1$ °C. Colonies were counted after 48–72 h incubation period. This procedure was repeated on peat soil from the three cutaway sites separately.

#### 2.3.2. Isolation and Enumeration of the Culturable Fungal Population

Serial dilutions ($10^{-1}$–$10^{-7}$), performed as described in Section 2.3.1., were plated on a half-strength Czapek–Dox agar medium. The agar medium was treated with streptomycin to inhibit the bacterial growth. The plates were incubated at $25 \pm 1$ °C. The colonies were counted after 2–3 weeks of the incubation period. The fungal colonies with different morphologies were subcultured onto new Czapek-Dox agar media. The fungal spores, conidia,

hyphae, and sporangia were observed under a light microscope. Primary identification of the fungi was performed according to the descriptions and mycological keys in Gams' and Bissett's (1998) Textbook of Fungi [14,15]. This procedure was repeated on peat soil from the three cutaway sites separately.

### 2.3.3. Isolation and Enumeration of Anaerobic Bacteria

The serial dilutions ($10^{-1}$–$10^{-7}$), as described in Section 2.3.1., were pour-plated onto anaerobic agar media. The plates were incubated inside the anaerobic jars with the Microbiology Anaerocult® A (a reagent for the generation of an anaerobic medium) in them. The colony counts were recorded after 2–3 weeks of incubation at $25 \pm 1$ °C conditions. Morphologically different colonies were subcultured for three consecutive times to obtain pure isolates. Gram staining was performed for the purified isolates. This procedure was repeated on peat soil samples from the three cutaway sites separately.

### 2.3.4. Isolation of Phosphate-Solubilising Bacteria (PSB)

The sterilised Pikovskaya medium was initially prepared without dextrose (*glucose*) to avoid the sugar caramelisation. The filter-sterilised *dextrose* solution was added to the sterilised medium separately. The dilution series ($10^{-1}$–$10^{-7}$) as described in Section 2.3.1. were spread plated on Pikovskaya agar. Control plates were set up by spreading 0.1 mL of sterile maximum recovery diluent onto Pikovskaya agar. The plates were incubated at $30 \pm 1$ °C for three weeks. Colonies with halozones were detected. The diameters were measured using a standard ruler. The reading error for a standard ruler with mm increments was $+/-0.1$ mm under optimal conditions. The uncertainty value was indicated for the diameter lengths in centimeters [16]. The *phosphate* solubilisation index (*PSI*) for each colony was calculated using the following equation.

Phosphate solubilising index (PSI) = {(Diameter of the halozone + colony) cm/(Diameter of the colony) cm}

### 2.3.5. Isolation of Actinomycetes

Serial dilutions ($10^{-1}$–$10^{-7}$), as described in Section 2.3.1., were plated onto starch casein agar. The plates were incubated at $25 \pm 1$ °C. Colonies with different morphologies were subcultured after 2–3 weeks of incubation. All the bacterial colonies with different morphologies were subjected to subculture conditions by three successive streak isolations in order to obtain pure cultures. Parallelly, Gram-negative and Gram-positive bacteria were differentiated by Gram's staining.

### 2.3.6. Glycerol Stock Preparation

A glycerol stock solution (200 mL) of a 40% (*v/v*) concentration was prepared by mixing glycerol (80 mL) with de-ionised water (120 mL). The stock solution was autoclaved. Pure isolates were obtained from total viable bacteria, aerobic bacteria, fungi, anaerobes, phosphate-solubilising bacteria and actinomycetes. Each pure bacterial isolate (single colony) was inoculated in nutrient broth overnight. Fungal colonies were inoculated in Czapek–Dox modified broth. Overnight pure culture (500 μL) was banked in a sterilised 40% (*v/v*) glycerol solution (500 μL) in sterile cryovials. They were gently mixed and labelled. Cryovials were stored at $-80$ °C.

### 2.4. Measuring the Soil pH

The pH values of each peat sample were measured using the soil survey standard test method. A 1:5 soil: water suspension (*w:v*) was prepared as follows: 10.0 g of peat was weighed into a clean duran bottle. A volume of 50 mL of de-ionised water was added into it. The soil suspension was mechanically shaken for 1 h at 15 rpm (LABWIT, ZWYR-D2402, Shanghai, China). The pH meter was calibrated prior to obtaining the readings. The electrode was immersed into the soil suspension and pH values were recorded.

### 2.5. Statistical Analysis

Prism version 9.4.0 Graph pad Software, Inc. was used to produce the graphs and perform the statistical analysis. All the samples were analysed in triplicate. Data were presented in a logarithmic scale and error bars of all figures were presented using the mean with standard deviation (SD). Multiple comparison analysis was performed using Tukey's test, unless otherwise stated. Statistical significance differences of the microbial population numbers were analysed using one-way analysis of variance (ANOVA) and two-way ANOVA with Tukey's post hoc test.

## 3. Results

### 3.1. Total Viable Bacterial (TVB) and the Aerobic Bacterial (AB) Population

The microbial numbers illustrated in this paper were based on a certain 'land use' type in peatlands. All the three sites described in this section were categorised under cutaway, rough and grazing. These three sites were located in different venues in the midlands. However, their vegetation and the 'land use' types were quite similar. Therefore, our study made an attempt to compare the different types of microbial groups in three different climatic changes and determine how they vary with each group of microorganisms. The average pH of peat soil in January, February and April was $4.15 \pm 0.03$, $6.03 \pm 0.05$ and $5.35 \pm 0.04$, respectively.

According to Figure 2a, a greater number of total viable bacteria ($1.33 \times 10^5$ CFU ml$^{-1}$) was recorded from the cutaway site at which the samples were collected in February. There was a statistically significant difference ($p \leq 0.0001$) between the numbers of total viable bacteria across the three sites on January, February and April (Figure 2a). The total bacterial number was $1.14 \times 10^3$ CFU ml$^{-1}$ in April. It was the lowest quantity among the three time points. This pattern was not observed for the aerobic bacteria. The numbers of aerobic bacteria were $1.71 \times 10^5$ CFU ml$^{-1}$ and $1.20 \times 10^5$ CFU ml$^{-1}$ in April and February, respectively (Figure 2b). However, there were slight differences in aerobic numbers across the three sites (Figure S1). According to Figure 2b, the comparison of TVB with AB populations indicates interesting findings. There was a statistical difference between cutaway samples collected from January and April in terms of TVB and AB populations ($p \leq 0.0001$). Furthermore, there was a statistical difference between the February sample to the rest of the cutaway sites ($p \leq 0.05$). Peat collected in January and April showed a higher AB population than the TVB. This phenomenon was reversed in February (Figure 2b). The ratio of AB to TVB was greater than 1.4 in January and April. It was less than 1 in February (Figure S2).

### 3.2. Morphology of Bacteria under Light Microscopy

The microscopic observation of pure bacterial isolates revealed most were predominantly Gram-positive rods arranged as chains. (Figure 3a) They could be an indication of *Bacillus* spp. Cells with central endospores and sub-terminal endospores (Figure 3b) were detected. The proportion of Gram-negative (Figure 3c) isolates was comparatively less than that of Gram-positive bacteria. Occasional Gram-negative short-rod (Figure 3c) cells were also detected.

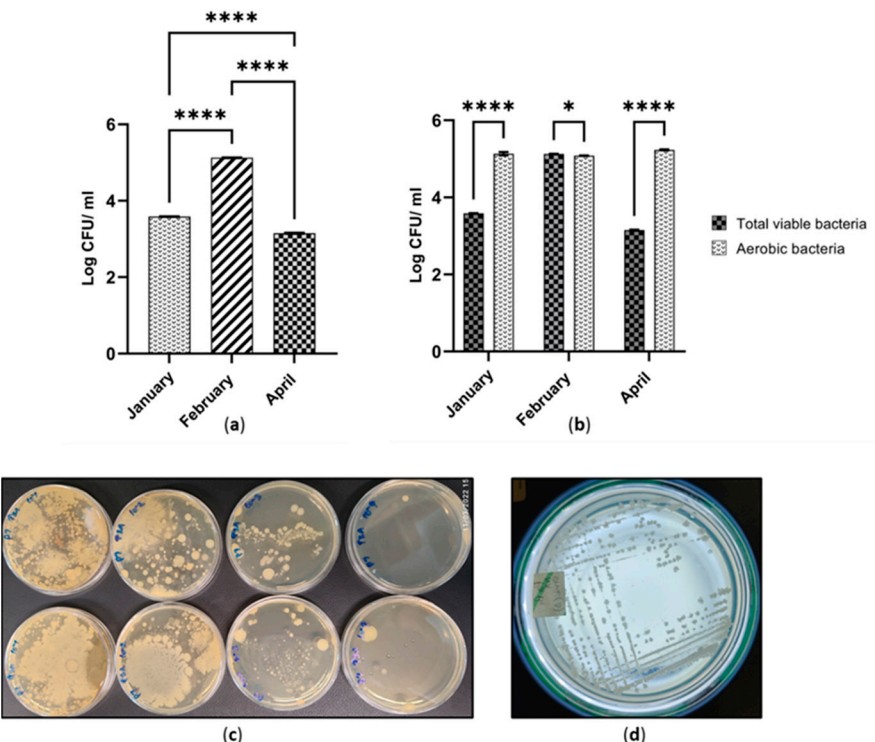

**Figure 2.** (**a**) Number of total viable bacteria across three cutaway sites cultured in three different time periods; (**b**) Comparison of total viable bacteria and aerobic bacteria; * $p \leq 0.05$; **** $p \leq 0.0001$; (**c**) Initial serially diluted pour plate isolation of TVB in cutaway peat collected in February; (**d**) Typical viable bacterial pure culture isolated in January.

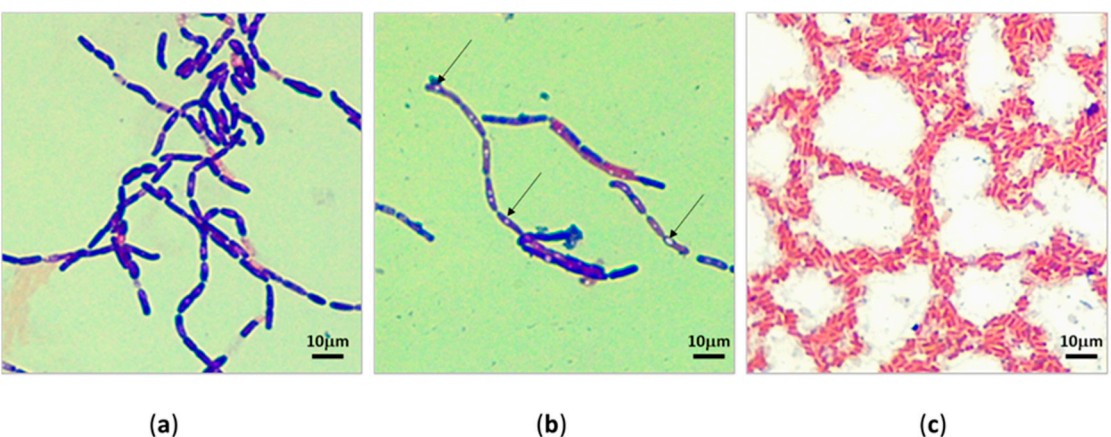

**Figure 3.** Light microscopy of subcultured bacterial isolates under oil immersion lens (10 × 100). (**a**) Gram-positive bacilli isolated in April cutaway site; (**b**) Gram-positive bacilli with sub-terminal endospores, the arrows point to the sub-terminal endospores; (**c**) Gram-negative short rods isolated in April cutaway site.

### 3.3. Variation of Fungal Population in Three Temporal Changes in Different Cutaway Sites

Numerous types of morphologically different fungal species were isolated from all three sites. The sub-cultured fungal isolates were phenotypically different in colour, shape, size and the colony formation. According to Figure 4a, the fruiting body of *Penicillium* spp. was observed under the light microscope. It was detected in all the three cutaway sites. The growth rate of *Penicillium* spp. was comparatively higher than that of the other fungal species. Colonies of *Penicillium* spp. appeared on the Czapek–Dox agar during the first five days of incubation (Figure 4b,c). However, most of the other fungal species required two weeks of incubation for colony development.

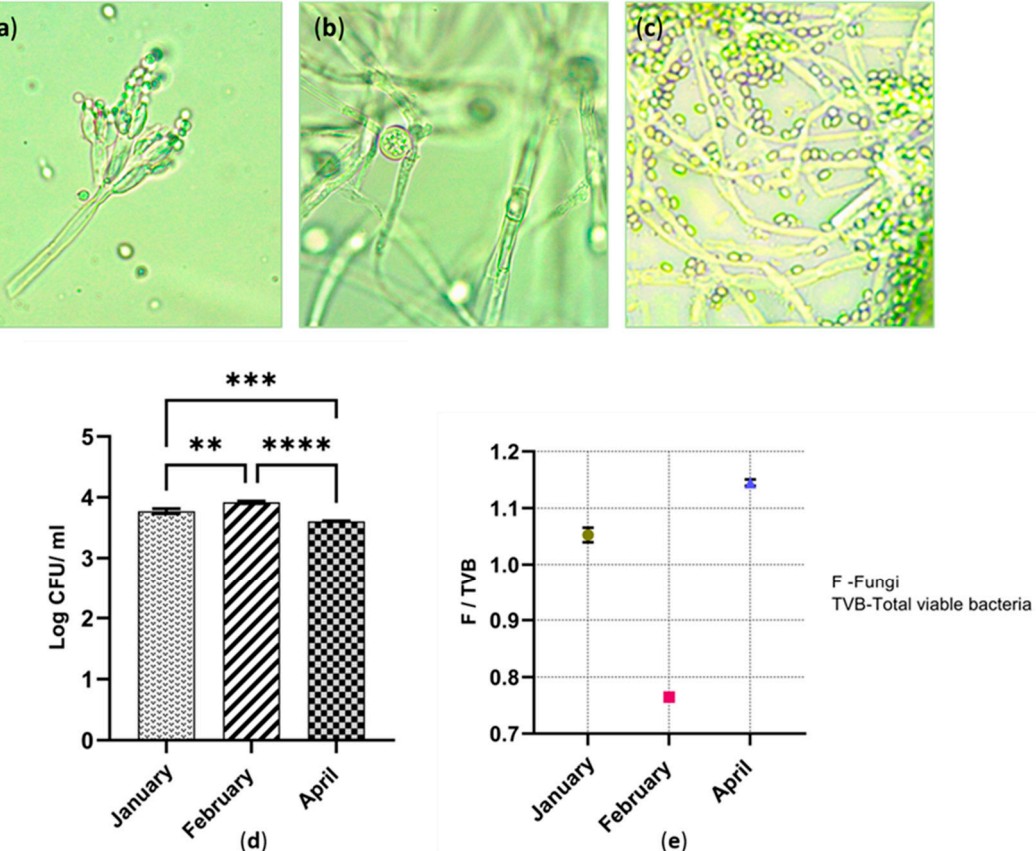

**Figure 4.** (**a**) Image of *Penicillium* spp. conidia-bearing structures under light microscope (10 × 100); (**b**) Image of spores inside sporangia under light microscope; (**c**) Image of *Trichoderma* spp. spores under light microscope (10 × 100); (**d**) Fungal population across the three cutaway sites; ** $p \leq 0.01$; *** $p \leq 0.001$; **** $p \leq 0.0001$; (**e**) The ratio of fungi to viable bacteria in each cutaway site.

The largest fungal population was found in February. April's sample denotes the smallest fungal population. Those values are statistically different ($p \leq 0.0001$). In accordance with the result shown in Figure 4d, the fungal populations significantly differ with each other during the three time periods. There was also a statistical difference between January and February ($p \leq 0.01$). Confirming Figure 4e, the fungi-to-viable bacteria (F/TVB) ratio was 1.05 and 1.14, in peat collected in January and April. The F/TVB ratio in February was significantly lower than that of the rest of the months. This value was less than 1, which was an indication of a higher proportion of viable bacteria (0.76).

### 3.4. Variation and Enumeration of Anaerobic Bacteria across the Three Sites

The maximum depth of peat collected was 15 cm. Therefore, the following anaerobic microbial numbers represent the peat layer above 15 cm of depth. A considerably green colour of colony development was achieved after 3 weeks of incubation. Gas formation was primarily detected at the bottom of the Petri dish. Some colonies developed at the bottom of the anaerobic agar layer. Other colonies were grown in the middle and on the surface of the agar medium. The abundance of anaerobic bacteria across the three study sites during tested January, February and April was illustrated in Figure 5a. Peat collected in January was the richest in anaerobic number ($1.14 \times 10^5$ CFU ml$^{-1}$). This value was significantly higher than the rest ($p \leq 0.0001$). On the contrary, there was no significant difference in anaerobic bacterial numbers between February and April ($p > 0.05$). Those values were lower than the number in January by approximately one order of magnitude. The predominant organisms were Gram-negative short rods in all the samples. Occasional

Gram-positive short rods were detected. Some isolates showed bizarre swellings in the middle of the bacterial cell. Some isolates exhibited terminal endospores.

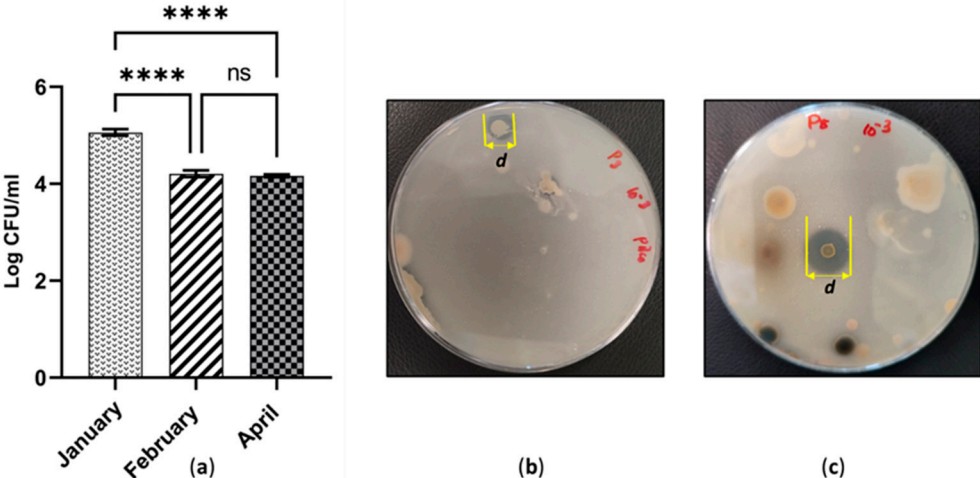

**Figure 5.** (**a**) Quantity of anaerobic population across cutaway sites in three different time periods; ns, not significant ($p > 0.05$); **** $p \leq 0.0001$; (**b**) Halozone development due to phosphate solubilisation by PSB isolated in January; (**c**) Halozone development due to phosphate solubilisation by PSB isolated in April.

*3.5. Phosphate-Solubilising Activity in Three Different Time Periods*

The level of phosphate-solubilising activity was determined using the Pikovskaya medium. It employs β-tri-calcium phosphate as the sole source of insoluble, inorganic phosphate. The concentration of the phosphate source and the pH of the medium were at constant values throughout the experiment. Therefore, the phosphate solubilisation index was interpreted for the phosphate solubility at pH 7.0. Interestingly, phosphate-solubilising bacteria were detected in all the three sites. The majority of the PSB isolates took 2–3 weeks to reach the maximum diameter of the halozone. According to Table 1. PSB, endowed a remarkable ability to solubilise calcium phosphate, were isolated in April (SI = 3.167 & 3.000, respectively). Comparatively, the worst ability to demonstrate phosphate solubilisation was recorded in January (SI = 1.25 & 1.26, respectively). It was noted that approximately 77.7 % of the PSB isolates demonstrates PSI values in the range of 1–2 and 28.57% above SI = 2.

**Table 1.** Phosphate solubilising index of PSB isolates after 3 weeks of incubation.

| PSB Isolate | Average Diameter of the Colony (cm) [1] | Average Diameter of the Colony + Halozone (cm) [1] | Phosphate Solubilisation Index (PSI) |
|---|---|---|---|
| P3S1 | $1.0 \pm 0.1$ | $1.25 \pm 0.1$ | 1.250 |
| P3S2 | $0.95 \pm 0.1$ | $1.2 \pm 0.1$ | 1.263 |
| P3S3 | $0.35 \pm 0.1$ | $0.5 \pm 0.1$ | 1.429 |
| P3S4 | $0.4 \pm 0.1$ | $0.525 \pm 0.1$ | 1.313 |
| P7S1 | $1.05 \pm 0.1$ | $1.425 \pm 0.1$ | 1.357 |
| P7S2 | $0.5 \pm 0.1$ | $0.65 \pm 0.1$ | 1.300 |
| P8S1 | $0.5 \pm 0.1$ | $1.5 \pm 0.1$ | 3.000 |
| P8S2 | $0.6 \pm 0.1$ | $1.9 \pm 0.1$ | 3.167 |
| P8S3 | $1.25 \pm 0.1$ | $2.0 \pm 0.1$ | 1.6 |

[1] The uncertainty value ($\pm$) was indicated with the average diameter measurements.

*3.6. Detection and Isolation of Actinomycetes*

Colony development on starch casein agar was achieved after 2–3 weeks of incubation. Colony development was comparatively slower than that of the typical bacteria. Many colonies exhibited the appearance of halozones. The halozones remained colourless

when the plates were treated with iodine solution. The typical characteristic colonies were 'pellet'-like and pigmented (grey, pink, maroon, yellow and orange). The presence of substrate mycelium and aerial mycelium was prominent. Gram-positive clusters of branched filamentous bacteria were detected under the light microscopye. They resembled filamentous fungi.

## 4. Discussion

Cutaway peatlands in general are former bogs [1]. They were converted into drained lands due to extensive use of land for peat extraction [2]. Everything, from the soil structure to microbial environment, is altered in cutaway peatlands due to 'land use'. The differences in the vegetation that occur during the formation of cutaway land have been studied in several attempts [1,17]. While it was proven to be an ideal approach to restoring the degraded peatland like cutaway sites, finding a solution from a microbiological point of view can be promising in setting up effective re-wetting schemes. Therefore, through this study baseline data of different microbiological groups were collected and analysed in three cutaway sites in the midlands over 3 months (2 seasons). Vegetation patterns in these lands showed some similarities. Peat soil from each site was collected in January, February and April. The two physiological factors considered are temperature and pH of the peat soil. The average temperature in peatlands during the three months were 6 °C, 7 °C and 10 °C, respectively. For the culture isolation of this study, both selective and general-purpose media were used. The study strongly intended to differentiate the isolation procedure for culturable microbial populations in peat. However, the aim of this study was to give optimal culture conditions in a laboratory and adapt the natural peat microorganisms to grow as regular microorganisms. The isolation of unculturable populations would require some adjustments such as adjusting the pH of the medium, as well as the temperature and preparation of media, using peat extracts and long incubation periods.

In February, both the total viable bacterial population (Figure 2a) and the fungal populations (Figure 4d) were greatest compared to the other two months. April recorded the lowest numbers of total viable bacteria (Figure 2a) and fungal population (Figure 4d). The temperatures of these two climates were around 7 °C and 10 °C (in February and April, respectively). While this temperature difference is apparent, the effect of temperature on the quantity of microbiota cannot be modelled without further data. However, the changing of total microbial numbers is not temperature-dependent. Physiochemical properties of peatlands over time can provide added value to explain the differences of microbial quantities over time. Both total viable bacteria (Figure 2a) and fungi (Figure 4d) reflect a similar pattern in terms of microbial quantities. The numbers after three months are significantly different from each other. This could be due to the differences in temperature, pH and the nutrient availability of the soil. Peat collected in January and April is much more acidic ($4.15 \pm 0.03$ and $5.35 \pm 0.04$ pH) than that collected in February ($6.03 \pm 0.05$ pH). The pH of the culture media was $7.0 \pm 0.2$, which is a neutral value. Initially, microorganisms thrived in acidic soil for a longer time. They face a difficulty in adapting to a neutral environment. However, the pH difference (6.03–7.0 pH) in peat and the culture media was lowest in February. The higher bacterial and fungal growth recorded in February could be due to this better chance of microbial adaptation to a neutral environment (7.0 pH). However, the numbers of aerobic bacteria do not resemble the same pattern as that displayed by the total viable bacteria. There is no significant difference between January's and February's aerobic population. Aerobes strongly depend on the availability of oxygen. If a higher proportion of peat soil was collected from the top layers during the sampling process, the possibility of isolating more aerobes is higher. In the current study, short incubation periods were employed to isolate aerobic and total bacteria. It was necessary to target the culturable organisms. A successful growth of aerobes was observed after 48 h of incubation [17].

A study conducted by Rebekka et al. (2007) examined the effect of vegetational succession on the fungal community and structure [18]. The effect was strongest for the peat soil collected from the surface horizons [18]. Differences in the fungal communities

have been proven by denaturing gradient gel electrophoresis fingerprinting [19]. Very few studies have actually detected such changes. Apart from the prokaryotic communities, eukaryotic organisms have also been addressed in the context of cutaway peatlands. In Ireland, some of the cutaway peatlands in the midlands have been converted into wetlands because of reflooding. Based on such sites, phytoplankton communities were assumed to be a great tool to monitor the chemical water quality [18]. Abundance of organisms such as dinoflagellates and blue-green algae have reflected the water quality in cutaway sites located in the midlands. These sites are rich in phosphorus and other minerals. Phytoplankton communities act as indicators to monitor the level of water quality in wetlands [20]. Prior to this study, Higgins and co-workers have analysed zooplankton species present in artificial lakes. These lakes were created on Irish cutaway peatlands. The study also revealed that the establishment of phytoplankton and protozoans is rapid when the cutaway lands are flooded [21]. The same principle can be applicable to the microbial community. One report shows that microbial abundance acts as an early indicator to observe the changes in soil quality. The particular study utilised peat microbes to measure soil carbon and nitrogen in peatlands [22].

The scientific literature states that microbial availability is greatly governed by factors such as temperature, moisture, oxygen availability and substrate concentration. However, root exudates of plant matter can influence the growth of some soil microbes [23]. In this study, most of the Gram-negative bacteria isolates were detected among the anaerobic bacteria (Figure 3c). One role of the Gram-negative bacteria includes the induction of the fresh carbon turnover [24]. Gram-positive isolates (Figure 3a,b) were mostly detected among the total viable bacteria across all the three sites. They have the potential to utilise recalcitrant carbon sources in peat soil [24]. A study carried out in Finnish cutaway sites analysed the microbial community structures using the phospholipid fatty analysis. The researchers observed statistically significant differences among the microbial community structures in Finnish soils. Peat, collected in the 10–15 cm layer, comprised higher proportions of Gram-positive bacteria than it did did of the Gram-negative bacteria [23]. Similar to our study, this study has also illustrated the fungi/bacteria (F/B) ratio with respect to different depths and sites [23]. The F/B ratios (Figure 4e) recorded in our study sites are comparatively higher, being similar to those found in previous studies [25]. The proportion of fungi is higher in drained peatlands than the bare peat [23]. This should be taken into consideration in the restoration process. The ratio of fungi: bacteria could be a fine indicator of effective restoration practices for bringing the drained lands into pristine condition. Cultivation and drainage can increase the abundance of fungi. However, peat extraction could drive this force to inhibit the growth of fungi [23]. However, when practices such as ecological succession occur on land, the fungal species can degrade the plant litter. Thereby, the F/B ratio gradually increases [23]. As per the results obtained in current study, the F/B ratios seen within three months are significantly different from each other (Figure 4e). To determine why, previous plant succession steps and the duration of peat extraction must be considered along with the temperature. A study conducted in China reveals that fungal communities are more sensitive than bacteria when they respond to drainage. The study concluded that the contribution of fungi is more significant than bacteria to developing overall microbial activity [26].

Another peatland study based in Spain hypothesised that microbial community structure can be governed by the temperature and moisture content in seasonal changes. As in many microbiological studies, PLFA profiling confirmed that the F/B ratio was very low. Additionally, changes in microbiota with the temperature were noticeable. The study did not detect any correlation between the peat botanical origin and the microbial community. However, factors like temperature and aeration of peat proved themselves to exert influence on microbial community composition [27]. An important microbial indicator like F/B (Figure 4e) is interpreted in the current study. It is considered to constitute a vital proxy for carbon transformations in peatlands [27]. As a pilot study, microbial parameters like

these would be undeniably important in collecting baseline data for the continuation of this project.

The majority of studies based on peatland microbiology address the dynamics of anaerobic microorganisms. The reason for this is that is most methanogenic bacteria are anaerobes. They act as the key reason for greenhouse gas emissions [17]. The type of vegetation is another vital element which is influential in the release of greenhouse gas emissions [28]. One underlying reason (in general) is that the lack of vegetation in abandoned cutaway peatlands encourages peat oxidation. This in turn increases $CO_2$ emissions [17]. Hypothetically, there is no exact single class of bacteria capable of breaking down the complex polymers in peat. A wide range of microorganisms are involved in the anaerobic degradation process in peatlands. Some microbes can produce methane gas in the anaerobic peat layer. The produced methane gas is utilised by other microorganisms residing in the aerobic peat layer [28]. Therefore, there is a rising necessity to explore the anaerobic microorganisms present in peat ecosystems. This is because it could build a platform to take measures to mitigate peatland drainage. A recent study, conducted by Urbanová and Bárta, showed the importance of studying the anaerobic community of drained peatlands. Based on the abundance of methanogenic community in pristine, re-wetted and drained sites, some vital conclusions have been drawn. Methanogenic abundance in drained site can reach an approximately pristine-like state after the re-wetting schemes has been applied. This proves that microorganisms are good indicators which can reflect successfully assist in the restoration of peatlands [11]. Therefore, the current study collected baseline data of abundance of anaerobic bacteria (Figure 5a) in three cutaway sites. The cutaway site in January records the highest number of anaerobic bacteria. The value is significantly higher ($p \leq 0.0001$) than that recorded in February and April. The initial growth of anaerobic bacteria leads to the release of $CO_2$ to the anaerobic medium. When the $CO_2$ dissolves, carbonic acid is produced. Carbonic acid can make anaerobic media slightly acidic, although they are slightly neutral initially. Considering the initial pH of the peat soil, anaerobes from the January sample could have a better opportunity to grow in a slightly more acidic medium than the rest. This could be one underlying reason for the higher numbers of anaerobes recorded in January. Comparatively low numbers of anaerobes were recorded in February and April. Without further information being available regarding any previous re-wetting steps, only narrow conclusions could be made from the initial enumerations of the anaerobic bacteria in our study. However, the comparative data of microbial quantities and the pure anaerobic isolates could be quite useful for drawing valid conclusions when this study is continued ahead of its pilot stage in the future. A study carried out in Finland has focused the methane-cycling microbial communities in Finnish peatlands. Based on the abundance of methanogens and methanotrophs, the authors concluded that it will at least take up to about 10 years to restore the forestry drained peatland. They predicted this result using the microbial indicators. Microbial indicators were isolated from the aerobic surface and the anaerobic surface of peat [29]. Several countries which are rich in northern peatlands have taken this approach. However, the Republic of Ireland has not entirely stepped into the application of microbial indicators to analyse the vegetation succession or the restoration succession after the re-wetting of cutaway peatlands.

Apart from the major, more characteristic microbial groups, actinomycetes and phosphate-solubilising organisms share a great responsibility in the decomposition process in peatlands. Knowledge concerning the role of these specialised microbial groups in Irish peatlands and other countries is limited. One such study carried out in North America studied the effect of the vacuum extraction of peat using microbes as biological indicators [30]. The difference between a natural peatland and a vacuum-extracted peatland was compared using microbial populations, including actinomycetes [30]. The were lower levels of the bacterial and fungal populations vacuum-extracted lands than in pristine-state lands [30]. The numbers of actinomycete populations were always lower than those of the typical bacteria, which is also consistent with our studies. However, it is not accurate to assume

that all the colonies grown on starch casein agar belong to the group of actinomycetes. Even saccharolytic organisms can grow on starch casein agar. Therefore, interpreting the colony counts on starch casein agar as a whole of actinomycetes number would not be entirely accurate. The population could represent both the saccharolytic and actinomycete organisms. In this present study, starch casein agar medium was used to isolate actinomycetes. The sole source of carbon used was starch. However, some actinomycetes are unable to utilise starch. They would not have been detected at any point. However, actinomycetes which utilise starch can release a significant amount of glucose (and other simple sugars) molecules to the medium. Thus, there is a possibility that actinomycetes, which cannot utilise starch but can rely on the glucose, will be released into the medium. The initial aim of the current study was to detect the presence of actinomycetes in these three cutaway sites using a standard selective medium. Starch casein agar had frequently been used in previous studies performed solely based on actinomycetes. The basis of selecting the isolation media was to isolate the preliminary yet culturable microbial population first. In our study, several distinct microbial groups have been primarily isolated in relation to their numbers. The authors took morphology into consideration as an initial step for the primary detection of actinomycetes. However, of course, if this study proceeds ahead, its pilot stage, a detailed characterisation of actinomycetes can be performed using the biochemical tests. The collective data of the actinomycetes can then be presented in a separate manuscript. Commonly isolated actinomycete species in previous research articles include *Micrococcus*, *Streptomyces* spp., and *Nocardia* [30]. The culture characteristics of the actinomycete colonies (Figure 6) isolated in each month are distinct. Further classification studies are yet to be conducted in order to identify the genus level. It should be emphasised that the majority of the soil actinobacteria are likely to thrive in aerobic conditions [31]. Therefore, it was the case in this study that, when the peat was taken in the below layers, it may have restricted the growth of strictly aerobic actinomycetes. Another study based on Finnish peatlands investigated the variation in actinobacterial populations among bogs. The composition of actinobacteria is correlated with the water table level. In conclusion, the writers state that when the water table level is altered, obvious changes will be detected in the actinobacterial community [31].

According to the results discussed here, the objectives of our study were achieved. This includes (i) the obtention of baseline microbial data on Irish peatlands, (ii) the comparison of microbial numbers (six different groups) in sites temporally and (iii) the possibility of adapting microbes as early indicators to monitor restoration progress. Owing to the importance of studying the microbial diversity in Irish peatlands, preliminary data were gathered to continue studies and initiate setting up promising re-wetting schemes in the future. The subsequent reclamation will have a significant impact on the Irish peatlands and will assist in minimising greenhouse gas emissions. The microbial parameters mentioned here will be valid in initiatives for the future peatland research to study Irish peatlands in depth. Preliminary indicators like F/B are tools to monitor peatland restoration over time. However, some possible future directions for peatland microbiology are worth mentioning here. The pure isolates obtained from the current research can be sequenced to identify possible indicators in each season. The isolation of unculturable populations can be performed in a follow-on study by setting up bio-reactors which exactly mimic peat environments. Data of unculturable microbial populations can be compared with the culturable population numbers and can be presented in the future. Moreover, since this pilot research does not anticipate being a conclusive study of peatland microbiology, sample sizes will be increased during future research. Combined isolation-based studies and metagenomics-based studies will provide a better picture of the total microbiota. Studies on microorganisms other than bacteria and fungi should be focused on the future since their role could be notable.

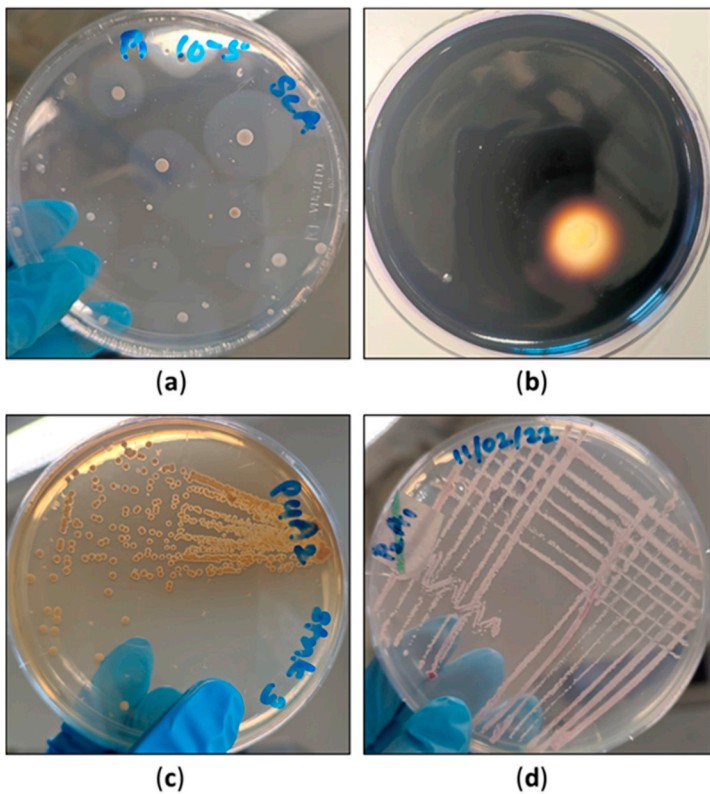

**Figure 6.** (**a**) Halozones developed due to starch hydrolysis by the bacteria grown on starch casein agar; (**b**) Visible clear zone around an actinomycete colony after the addition of iodine; (**c**) Pigmented actinomycete isolate obtained from the cutaway site in April; (**d**) Pigmented actinomycete isolate obtained from the cutaway site in January.

## 5. Conclusions

In accordance with the results obtained here, there is an overall statistically significant difference between the microbial numbers (TVB, Aerobes, fungi and anaerobes) across three cutaway sites. Among the three time points, the total viable bacterial and fungal numbers were highest in February when it is the coldest. The numbers of total viable bacteria and fungi were lowest in April when it is comparatively warmer. There is a statistically significant difference among the microbial numbers in three months. The difference in temperature and pH might be the reason for this significant variation in microbial number. *Penicillium* spp. and *Trichoderma* spp. are common in all the sites. The highest phosphate solubilisation index values are recorded from the cutaway peat collected in April (SI = 3.167 & 3.000). In view of the importance of studying the microbial diversity in Irish peatlands, preliminary data are generated while fulfilling the aims and objectives. This project also gives valuable future directions which will pave the way to the establishment of promising re-wetting schemes in order to reduce greenhouse gas emissions.

**Supplementary Materials:** The following supporting information can be downloaded at: https://www.mdpi.com/article/10.3390/applmicrobiol3010021/s1. Figure S1: Number of aerobic bacteria across three cutaway sites cultured in three different time periods, Figure S2: The ratio of aerobic bacteria to total viable bacteria in cutaway bogs in three time periods.

**Author Contributions:** Conceptualization, G.N.A.; methodology, G.N.A.; software, G.N.A.; validation, G.N.A.; formal analysis, G.N.A.; investigation, M.G. resources, M.G. and F.T.; data curation, G.N.A.; writing—original draft preparation, G.N.A.; writing—review and editing, T.B.; M.G. and F.T.; visualization, G.N.A.; supervision, F.T. and M.G.; project administration, M.G. and F.T.; funding acquisition, F.T. and M.G. All authors have read and agreed to the published version of the manuscript.

**Funding:** The project is funded by the European Innovation Partnerships from The Department of Agriculture, Food and the Marine (DAFM).

**Data Availability Statement:** Not applicable.

**Conflicts of Interest:** The authors declare no conflict of interest.

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
