# Peer review of "Do the Culturable Microbial Groups Present in Cutaway Bogs Change According to Temporal Variation? Pilot Study Based on the Midlands in the Republic of Ireland"

_2673-8007, doi:10.3390/applmicrobiol3010021_

Round 1

Reviewer 1 Report

This paper study the property of peat during three different climatic periods. I suggest that this paper can be revised after major revision. The authors should reply the following comments before this paper can be published.

Firstly, the English should be polished again.

Secondly, the test samples should be added. The conclusions drawn from current samples are not enough.

Thirdly, I think that the culture cycle is not enough. The authors should prolong the culture cycle.

Fourthly, I suggest that the authors should added more comparisons with your conclusions.

Reviewer 2 Report

This manuscript aims to describe observations as to how temporal variation results in changes in microbial culturable diversity of cutaway peatlands in Ireland. As with any study based on laboratory-based cultivation/isolation of microorganisms, only a limited snapshot of what is taking place can be observed. Therefore, there are specific aspects the authors need to address/consider:

1. Only selected isolation media and isolation conditions were used in this study, which would've limited the number and type of microorganisms isolated. To increase the chances of isolating peat-inhabiting microorganisms, it is recommended to mimic the environmental conditions as much as possible during isolation, e.g., adjusting the pH to the peat pH, incubation of the isolation plates at the average temperature of the peatland, preparing agar media using peat extracts, etc. It is unclear as to why this approach was not taken in this study, especially since it focused solely on an isolation approach.

2. Since limited information is provided through the use of a cultivation-based study, the true value of the study is not clear from the text. Please provide a clearer contextualisation of the data presented here. How will what you have learnt contribute towards the potential rehabilitation of the peatlands? Clearly highlight the value of taking this approach to answer your research question.

3. In Table 1, a standard deviation is indicated for the various readings. However, in the methodology section, it is indicated that the PSB plates were used in the isolation of bacteria and there is no mention of inoculating pure cultures into PSB agar. The methodology section should be detailed enough and clearly linked to the results reported in the study.

4. What was the basis for the selection of isolation media? For example, many actinomycetes are unable to utilise starch as a carbon source, and would therefore not have been detected on the isolation agar used in this study. In addition, actinomycetes represent a vast group of bacteria whose morphology varies quite a bit. Selection of these bacteria based on morphology would also mean missing out on certain types of actinomycetes, making the approach used in this study insufficient.

5. It is difficult to see how any comparisons can be made on the changes in microbial composition over the different months. The methodology employed is insufficient to provide deep insights. Slight changes in isolation techniques/media used can result in different outcomes. Changes in the physicochemical properties of the peatlands over time, may have provided added value to the results presented here, serving as a basis for the references made to the effect of the environmental conditions driving microbial populations. It is highly recommended to include this information, especially pH and temperature.

Minor aspects:

1. Figure 1b: which of the three peatlands is represented here?

2. Page 3: Why was such a short incubation period used for the isolation of aerobic bacteria? What was the pH of the isolation media?

3. Page 4: Clearly indicate how the Phosphate Solubilizing Index (SI) was calculated, and clearly indicate what is meant by 'control plates'.

4. Pages 5 and 6, Figure 4: All bacterial and fungal genus names should be written in italics, while 'spp.' should not be written in italics (applies to the whole manuscript).

5. Figure 3: Indicate the magnification under which the bacteria was visualised.

6. Figure 5: 'Halazone' should be 'halozone'.

7. Language editing is required.

Reviewer 3 Report

The article is devoted to an important environmental problem, namely cut-away peatland ecosystems.  In this study, the authors compared peat microbiota across the cut-away peatland sites in the three different climatic periods. In their work, the authors used classical traditional methods for quantitative accounting of cultivated microorganisms. It should be noted that in modern studies, such methods of studying soil microbiota are extremely rare. To date, molecular research methods are the most informative for studying the microbial diversity of soils. However, for a pilot preliminary study, the use of classical Nutrient mediums is quite acceptable.

Obviously, in order to develop recommendations for the restoration of the Irish peatlands, it is necessary to conduct more in-depth and expanded studies on the study of the microbial diversity of the studied soils.

A few points:

1.     It is a well-known fact that the number of microorganisms in soils strongly depends on moisture and temperature. What was the temperature and moisture of peat soil samples collected in three different climatic periods (January, February and April). There are only data on the temperature of February and April.

2.     It is necessary to indicate a reference to the key used for the preliminary identification of fungi.

3.     In figure 2в, it is also necessary to present the statistical significance between the number of aerobes in January, February and April.

4.     There are no quantitative data on actinomycetes. Why?

5.     In my opinion, in the discussion, the authors paid very little attention to the interpretation of their own data.

6.     There is no reference to Rebekka 2007. Reference 20 is missing in the text.

Round 2

Reviewer 1 Report

This paper can be accepted.

Author Response

We thank the reviewer  for this expert comments and suggestion given throughout the reviewing process. It really improved the quality of our manuscript.

Reviewer 2 Report

The authors have made extensive changes to the manuscript. However, there are still certain aspects the authors need to address/change.

The manuscript requires extensive language editing – here are just a few examples, but there are many more:

1.     In the title, the term ‘cultivable’ (=arable, tillable) is typically used in reference to crops, whereas for microorganisms, the term ‘culturable’ (=can be cultured) would be more correct.

2.     Abstract, lines 13-15: Poor sentence structure. I would recommend changing the sentences to ‘Cutaway peatlands in the midlands of the Republic of Ireland are rarely the focus of scientific studies. Due to peat extraction, the soil quality and related microenvironment is severely impacted and returning them to a ‘near natural state’ would require greater insights into this ecological niche.’

3.     Line 16: ‘…the microbiology of three…’

4.     Line 17: According to this sentence, you are trying to culture peat on different culture media. The sentence needs to be adjusted so that it refers to the isolation of microorganisms ‘on a range of specific and non-specific culture media’.

5.     Line 18: ‘gram staining’ should be written with as ‘Gram staining’. The staining technique was named after the person who invented it and should therefore be written with a capital ‘G’.

6.     Line 23: change ‘in three months’ to ‘across the three months focused on in this study’.

Other aspects:

·      At the end of the abstract, it is custom to add a line indicating what the conclusion was of the study. Please include a sentence here.

·      Line 71 and throughout the manuscript: Please see comment above regarding the terms, ‘cultivable’ vs ‘culturable’.

·      Line 94: Please provide the year of sampling.

·      Lines 101-102: It is unclear what the sentence here is meant to infer – were there only a few vegetation types present of which the mosses and sedges dominated or were there only a few Sphagnum mosses and sedges present?

·      Line 122: Change ‘prepared’ to ‘as described in section 2.3.1’.

·      Line 127 and throughout the manuscript: Avoid the use of the colloquial term ‘done’. It is more scientific to use the term ‘performed’.

·      Line 132: You can’t ‘prepare’ something in a section; you describe it in that section. Please also correct the spelling mistake ‘pour pated’ – should be ‘pour plated’.

·      Line 138: The section numbering is incorrect, and the text should not be in italics.

·      Line 157: Please indicate what the % solution refers to – w/v or v/v?

·      Lines 181: Surely there should be more than three pH readings – didn’t you sample from three different cutaway sites in January, February, and April (three different sites, three different months)? And if all experiments were performed in triplicate as indicated in section 2.5, there should be standard deviations that should be reported for these pH readings.

·      Line 284: Does this mean that the readings were based on a single test per isolate? In section 2.5 it is indicated that all tests were performed in triplicate…

·      Lines 499-502: Even if you try to mimic the peat environment during the isolation of microorganisms, there will still be a large proportion you will not be able to culture. There needs to be a combined isolation-based study and a metagenomics-based study in order to get a better picture of the total microbial population present. The role of microorganisms other than bacteria and fungi are also very important and should also be taken into consideration in future studies.

Author Response

We would like to thank the reviewer for these expert comments and suggestions given throughout this review process. It really improved the quality of the manuscript. We appreciate your feedback.
